# Fgf signalling triggers an intrinsic mesodermal timer that determines the duration of limb patterning

Sofia Sedas Perez [1,5], Caitlin McQueen [1,4,5], Holly Stainton[1], Joseph Pickering[1], Kavitha Chinnaiya[1], Patricia Saiz-Lopez[2,3], Marysia Placzek [1], Maria A. Ros [2,3] & Matthew Towers [1] ✉

Complex signalling between the apical ectodermal ridge (AER - a thickening of the distal epithelium) and the mesoderm controls limb patterning along the proximo-distal axis (humerus to digits). However, the essential in vivo requirement for AER-Fgf signalling makes it difficult to understand the exact roles that it fulfils. To overcome this barrier, we developed an amenable ex vivo chick wing tissue explant system that faithfully replicates in vivo parameters. Using inhibition experiments and RNA-sequencing, we identify a transient role for Fgfs in triggering the distal patterning phase. Fgfs are then dispensable for the maintenance of an intrinsic mesodermal transcriptome, which controls proliferation/differentiation timing and the duration of patterning. We also uncover additional roles for Fgf signalling in maintaining AER-related gene expression and in suppressing myogenesis. We describe a simple logic for limb patterning duration, which is potentially applicable to other systems, including the main body axis.

Much is known about how developing tissues and organs are spatially patterned. However, we have less knowledge about how the rate and/or duration of patterning events are determined both within and between different species - often referred to as developmental timing[1,2]. This is important because patterning events need to be temporally coordinated and larger species tend to develop at much slower rates than smaller species[3]. The developing limb is an excellent system for understanding how the duration of patterning is determined as we have extensive knowledge of the underlying mechanism, which is composed of early and late patterning phases[4] (Fig. 1a). The early proximal patterning phase (red – Hamburger Hamilton stage HH18-22 in the chick wing) involves the stepwise activation of *Hoxa/d10/11* genes in proliferative distal mesoderm cells (dark blue circles in Fig. 1a), and requires a proximal signal - generally considered to be retinoic acid (RA) - emanating from the main body of the embryo[5–10]. Hoxa/d10/11 specify cells with proximal positional values, which direct

their development into the stylopod and the zeugopod[11,12] (humerus and the ulna/radius - Fig. 1a). The depletion of RA is influenced by growth of the limb away from the body and by the degradation enzyme, Cyp26b1, which is transcriptionally induced by Fibroblast growth factor (Fgf) signals from the apical ectodermal ridge[13,14] (AER-Fgfs – the AER is a thickening of the distal epithelium – Fig. 1a). In different avian species (quail, chick and turkey), the duration of the early proximal patterning phase varies and takes between 12 and 30 h[15]. The loss of RA signalling in the distal part of the limb then allows the initiation of the late distal patterning phase (light blue in Fig. 1a – HH22-29 in the chick wing), coinciding with *Hoxa/d13* gene activation[9,16], which specifies the positional values of the autopod[17] (carpals, metacarpals and digit phalanges). In the quail, chick and turkey, the late distal patterning phase runs for a similar duration and takes between 48 and 54 h[15]. Prolonging RA signalling in quail and chick wings extends the early proximal patterning phase, and because the

[1]School of Biosciences, University of Sheffield, Western Bank, Sheffield S10 2TN, UK. [2]Instituto de Biomedicina y Biotecnología de Cantabria, IBBTEC (CSIC-Universidad de Cantabria), 39011 Santander, Spain. [3]Departamento de Anatomía y Biología Celular Facultad de Medicina, Universidad de Cantabria, 39011 Santander, Spain. [4]Present address: Chester Medical School, Chester CH2 1BR, UK. [5]These authors contributed equally: Sofia Sedas Perez, Caitlin McQueen. ✉e-mail: m.towers@sheffield.ac.uk

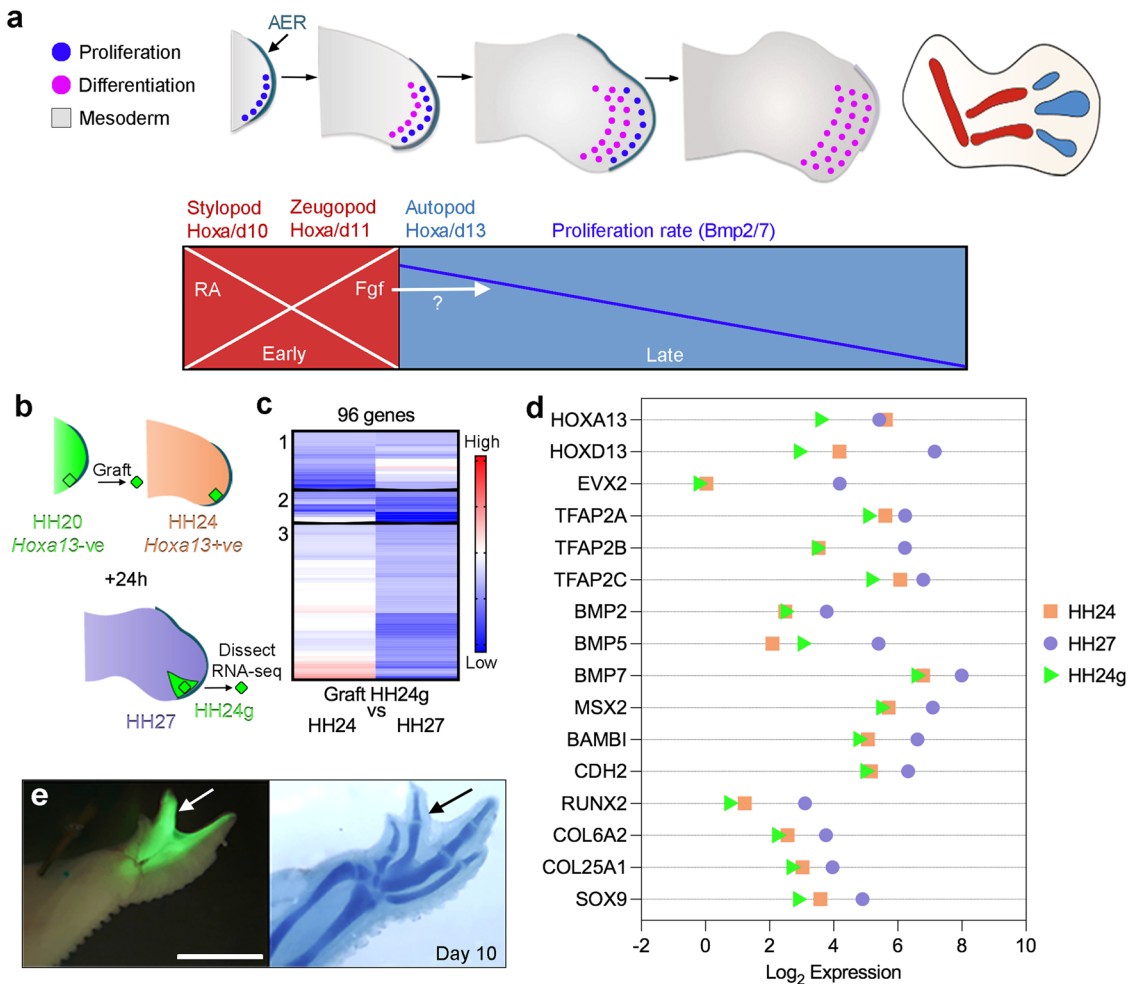

**Fig. 1 | Proximo-distal patterning of the chick wing and the intrinsic mesoderm transcriptome. a** A population of undifferentiated proliferative mesoderm cells (dark blue circles) is maintained beneath the apical ectodermal ridge (AER) at the distal tip of the wing - cells displaced proximally differentiate (pink circles) into proximal structures and then distal structures as outgrowth proceeds (stylopod - humerus, zeugopod –ulna/radius and then autopod – metacarpals, carpals and digit phalanges). Early proximal patterning phase: retinoic acid (RA) signalling from the flank is opposed by Fgf signalling from the AER (AER-Fgfs), which permits the mesodermal expression of *Hoxa/d10* and *Hoxa/d11* genes that specify the positional values of the stylopod and the zeugopod. Late distal patterning phase: clearance of RA is suggested to create a permissive environment for the mesodermal activation of *Hoxa/d13* genes that specify the positional values of the autopod. An intrinsic mesodermal Bmp2/7-dependent timer is proposed to control the duration of the patterning phase, and the involvement of AER-Fgfs is an open question. **b** Procedure to find the intrinsic mesoderm transcriptome: $150^3$ μm blocks of GFP-expressing (green – *Hoxa13*-ve) HH20 chick wing bud distal mesoderm was denuded of ectoderm and grafted under the AER of wild type HH24 buds (orange - *Hoxa13*+ve), incubated for 24 h until the host stage is HH27 (purple) and the graft stage is HH24 (HH24g - green), and then the distal-most $150^3$ μm of the graft was subjected to RNA-sequencing. **c** Clustering of RNA-sequencing data across pairwise contrasts with the log₂-fold change degree of gene expression indicated by the colour (red: higher, blue: lower). **d** Plot showing expression levels of *Hoxa13, Hoxd13, Evx2, Tfapa, Tfapb, Tfapc, Bmp2, Bmp5, Bmp7, Msx2, Bambi, N-cadherin (Cdh2), Runx2, Col6a2, Col25a1* and *Sox9* as normalised log₂ values of the RNA-sequencing read-count intensities. **e** Grafts of HH20 mesoderm made to HH24 buds often give rise to complete digits as shown at day 10 ($n = 12/35$). Scale bar: 1 mm. Source data are provided as a Source Data file.

late distal phase remains unchanged, the duration of patterning is then equivalent to that of chick and turkey wings, respectively[15]. It is likely that the degradation of RA and/or its transcriptional effectors (Meis1/2) is responsible for species' differences in patterning duration.

During the late distal patterning phase, the AER and underlying mesoderm continue to crosstalk via reciprocal signalling (epithelial-mesodermal or e-m signalling)[18]. The AER is an essential structure and its removal in the chick limb bud truncates outgrowth[19,20], which can be rescued by a bead soaked in Fgf protein implanted into the distal mesoderm[21,22]. In turn, the mesoderm maintains AER-Fgfs via the production of Fgf10[23] and of the Bmp (Bone morphogenetic protein) antagonist, Gremlin1 (Grem1)[24,25]. Embryological experiments performed on the chick wing suggest that an intrinsic Bmp2/7-dependent mesodermal proliferation timer controls the duration of the late distal patterning phase, which ends when all cells have differentiated[16,26,27]

(pink circles in Fig. 1a). The role of AER-Fgfs in this model is to maintain mesoderm proliferation as permissive factors. By contrast, mouse genetics support an instructive role for AER-Fgfs[28–33], although this is difficult to verify because of their essential requirement during limb development[30] (Fig. 1a).

Here, we use heterochronic grafting techniques coupled with RNA-sequencing to further decipher the extent to which the late distal patterning phase is directly controlled by extrinsic or intrinsic mechanisms in the chick wing mesoderm. We then develop an experimentally amenable tissue explant system to dissect the roles of Fgfs. We provide evidence that Fgfs are required for initiating the late distal patterning phase by activating *Hox13* genes. However, they are not required thereafter for intrinsic mesodermal proliferation/differentiation timing. We also reveal additional roles for Fgfs in maintaining AER-related gene expression and in suppressing myogenesis.

## Results

### The intrinsic chick wing mesoderm transcriptome

To understand the extent to which the late distal patterning phase is intrinsically regulated in the chick wing mesoderm we identified the underlying transcriptome. Previous heterochronic grafting techniques revealed that *Hoxa13* is intrinsically activated in chick wing mesoderm according to the age of the donor tissue, and that proliferation parameters, as well as cell adhesion properties are also maintained intrinsically[16]. To identify the intrinsic mesodermal transcriptome, we used the same experimental approach and applied RNA-sequencing. For the donor, we used transgenic chick embryos that ubiquitously and constitutively express Green Fluorescence Protein (GFP+ve) and grafted 150 μm³ blocks of HH20 (Hamburger Hamilton) distal mesoderm beneath the AER of wild type HH24 host wing buds (which are older by 24 h), and allowed them to develop for 24 h, so that the developmental stage of the graft is HH24g (graft), and the host is HH27 (Fig. 1b).

After HH20 grafts were made to HH24 wings and left for 24 h, we performed RNA-sequencing on 150 μm³ of the distal-most GFP+ve grafted mesoderm (HH24g, taking care not to dissect host GFP-ve tissue), and for controls, equivalent (non-grafted) HH27 tissue in the contralateral host wing bud, as well as HH24 tissue from the wing buds of different embryos (Fig. 1b). This identified 303 differentially expressed genes: 73 between HH24 and HH24g datasets and 230 between HH27 and HH24g datasets (>2-fold difference with an adjusted *p*-value of <0.05 – Supplementary data 1 and Supplementary data 2). We then used hierarchical clustering analyses to identify those genes that are intrinsically activated in the grafts like *Hoxa13*. Based on our previous analyses, we expected intrinsically activated genes to be expressed in the grafts (HH24g) at lower levels when compared with host levels (HH27), consistent with their later activation[16]. Three out of nine clusters contain genes that behave in this manner: cluster 1 (23 genes), cluster 2 (12 genes) and cluster 3 (61 genes (Fig. 1c; Supplementary Figs. 1–3). Expression was generally reduced in the grafts (HH24g) compared with donor levels (HH24), suggesting that the grafting procedure slightly delays gene activation as we noticed previously[16]. Nonetheless, most of the genes show similar trends in plots of the read-counts of the RNA-sequencing data, indicating that expression in HH24g samples (green triangles) is generally lower than, or equivalent to HH24 samples (orange squares), but substantially lower than HH27 samples (purple circles - Fig. 1d). The identification of *Hoxa13* confirmed the results of our previous study[16], and is the only gene characterised in limb development that is found in cluster 1. *Hoxd13* and *Evx2*, which are co-regulated syntenic genes, are found adjacent to each other in cluster 2. Many genes that are involved in limb development are present in cluster 3, including three that encode members of the Ap2 family of transcription factors, Tfap2a, Tfap2b and Tfap2c, which have been implicated in maintaining the undifferentiated state of the distal mesoderm[34,35]. In addition, several genes encoding members of the Bmp pathway are found, including the Bmp2, Bmp5 and Bmp7 ligands, and their downstream transcriptional effector, Msx2. *Bambi* is also present, which encodes a pseudoreceptor that limits the range of Bmp signalling in chick wing distal mesoderm[36]. These findings support those of our earlier study in which we revealed that BMP signalling regulates an intrinsic proliferation timer in the chick wing mesoderm[27]. Additional genes of interest in cluster 3 include *N-cadherin*, which encodes a molecule that mediates cell adhesion along the proximo-distal axis[37], and differentiation regulators, including *Col25a1* and *Col6a2* (connective tissue), *Sox9* (cartilage) and *Runx2* (bone). This is consistent with the finding that grafts of HH20 distal mesoderm made to HH24 buds often give rise to a complete digit, thus demonstrating their intrinsic differentiation potential (Fig. 1e). Genes that could act downstream of Bmp signalling in progressively suppressing G1- to S-phase entry are not represented in the intrinsic transcriptome, possibly because core cell cycle regulators are predominantly controlled at the post-translational level. However, a likely candidate is the Cyclin D inhibitor, $p57^{kip2}$ (CDKN1C)[38], which is expressed in the mesoderm of chick wing buds specifically during the late distal patterning phase between HH23 and HH29 (Supplementary Fig. 4a). Moreover, $p57^{kip2}$ is transcriptionally regulated by Bmp signals: it is induced/upregulated or repressed/downregulated 24 h after beads soaked in Bmp2, or the Bmp signalling inhibitor, Noggin, are implanted into the distal mesoderm of HH24 wings, respectively (Supplementary Fig. 4b–e). These data reveal the transcriptome involved in intrinsic mesoderm development.

### Chick wing explants proliferate with in vivo rates

We did not identify genes associated with Fgf signalling in the intrinsic mesoderm transcriptome (Supplementary Figs. 1–3). However, it is possible that Fgf signalling permissively maintains intrinsic mesodermal gene expression. This is difficult to validate experimentally because of the essential requirement for the AER and for Fgfs during in vivo limb development[19,20,30]. We investigated whether a tissue explant system could circumvent the essential functions of the AER and Fgfs, by culturing the posterior-distal third of HH20 chick wing buds in Matrigel (Supplementary Fig. 5a). To begin with, we included the Sonic hedgehog (Shh)-producing polarising region (ZPA) – a developmental organiser located in the posterior-distal mesoderm – because it makes an essential reciprocal signalling loop with AER-Fgfs[39,40]. We determined whether in vivo proliferation parameters are maintained in explants using flow cytometry, which gives an accurate stage-specific read-out of cell cycle rate[16,27] (percentage of cells in G1-phase). The analyses reveal that there is no significant difference in proliferation rates between in vivo tissue and explanted tissue at 24 and 48 h (Supplementary Fig. 5b).

### Explants closely maintain in vivo gene expression parameters

Having developed a chick wing tissue explant system that maintains in vivo proliferation timing, we determined gene expression parameters using the multiplex RNA-fluorescence in situ hybridisation (FISH) with amplification by hybridisation chain reaction (HCR) technique followed by light-sheet microscopy. Over time, *Meis1* and *Hoxa11* expression are excluded from the distal mesoderm and become restricted to proximal regions of both wing buds and explants (Fig. 2a–d, note, distal is to the top of the panels and posterior is to the right-hand side). In addition, *Hoxa13*, *Hoxd13*, *Shh*, *Bmp2*, *Sox9* and *Runx2* are expressed in distal regions of both wing buds and explants (Fig. 2e–p). In explants, there is a slight delay of approximately 6–12 h in the clearance of *Meis1* and *Hoxa11* from distal regions, and in the activation of *Hoxa13* and *Runx2*, which is likely to be caused by acclimatisation to the culturing conditions (Fig. 2a–d, e–f, o–p). Although digit condensations are observed in wing buds as indicated by *Sox9* and *Runx2* expression at 72 h (Fig. 2m, o), they are not as pronounced in explants (arrowheads - Fig. 2n, p). Taken together, these findings indicate that explants closely replicate in vivo gene expression parameters.

### Explants without an AER proliferate with in vivo rates

To begin to test the requirement for Fgfs in the late distal patterning phase in chick wing explants, we removed the AER at 0 h, as shown by the absence of *Fgf8* expression at 24 h (Supplementary Fig. 6a, b). In addition, *Shh* expression is also undetectable, consistent with a role for the AER in maintaining the activity of the polarising region[39–41] (Supplementary Fig. 6a, b). The removal of the AER dramatically attenuates Fgf signalling, as determined by a downstream readout, *Mkp3* (also known as *Dusp6* and *Pyst1*)[42], which is either absent or expressed at very low levels at 24 h (Supplementary Fig. 6c, d). This finding could indicate that expression and/or signalling by mesodermal Fgfs is also affected by the removal of the AER - in particular, Fgf10, that reciprocally maintains *Fgf8* expression[23]. However, *Fgf10* is expressed at

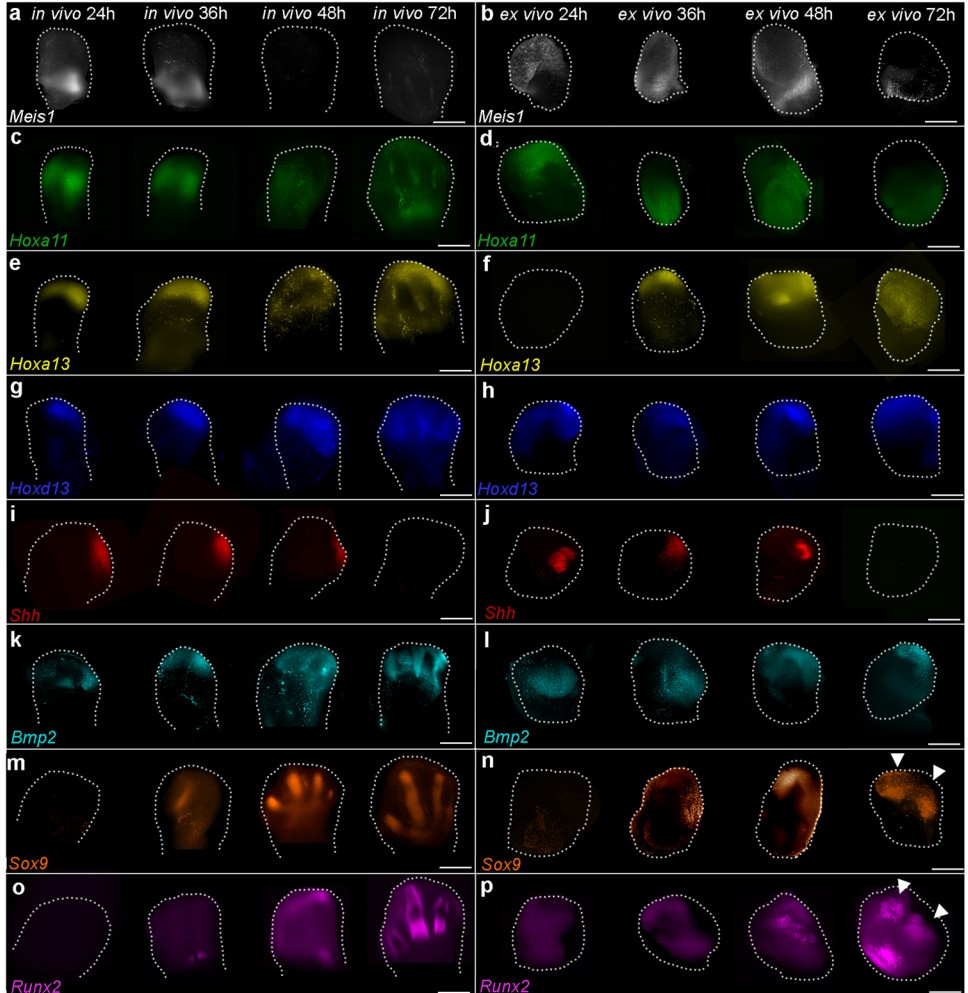

**Fig. 2 | Gene expression timing in explants. a, b** *Meis1*, (**c, d**) *Hoxa11*, (**e, f**) *Hoxa13*, (**g, h**) *Hoxd13*, (**i, j**) *Shh*, (**k, l**) *Bmp2*, (**m, n**) *Sox9* and (**o, p**) *Runx2* expression in wing buds (in vivo) and explants (ex vivo) over 72 h shown by HCR in situ hybridisation (*n* = >3 in all cases - distal is the top of the panels; posterior is the right). There is a delay in the clearance of *Meis1* (**a, b**) and *Hoxa11* (**c, d**) from the distal part of explants and in the activation of *Hoxa13* (**e, f**) and *Runx2* (**o, p**). Digit condensations are marked by *Sox9* and *Runx2* expression in (**m, o**) wing buds and (**n, p**) explants (arrowheads). Scale bars for wing buds − 500 µM and explants − 200 µM.

reduced levels at 24 h, thus indicating that it is regulated independently of Fgf8, and that *Mkp3* expression relies more on AER-Fgfs than on mesodermal-Fgfs (Supplementary Fig. 6e, f).

Despite losing the activity of both the AER and the polarising region, flow cytometric analyses reveal no significant difference in the percentage of G1-phase cells in explants compared with controls at 24 and 48 h (Fig. 3a). Additionally, EdU labelling demonstrates no significant difference in the percentage of cells that are transiting through S-phase at 24 and 48 h in controls (Fig. 3c, d, f) compared with explants cultured without an AER (Fig. 3c, e, f). Furthermore, although explants without an AER appear to have a smaller surface area, it is not significantly different to explants that have an intact AER (Fig. 3g). However, compared with controls, in which the posterior necrotic zone can be detected by lysotracker staining at 24 and 48 h (Fig. 3h, j, l), a significant 2.91- and 2.71-fold increase in apoptosis is detected in explants without the AER at 24 and 48 h, respectively (Fig. 3i, k, l). These findings reveal that the AER and the polarising region are dispensable for mesodermal proliferation timing in explants.

**Explants proliferate normally without Fgfs**
To inhibit Fgf signalling in chick wing explants in a more direct way than by removing the AER, we used a pharmacological approach, by applying the Fgf receptor inhibitor, SU5402, to the culture medium at 0 h. *Mkp3* is observed at low/background levels in explants treated

with SU5402 at 24 h (Supplementary Fig. 7a, b). In addition, SU5402 application reduces *Fgf10* expression levels at 24 h, indicating that it is transcriptionally regulated, at least in part, independently of Fgf signalling (Supplementary Fig. 7c, d). To determine the proliferative requirement for Fgf signalling, we performed flow cytometry and found no significant difference in the percentage of G1-phase cells in explants treated with or without SU5402 for 24 and 48 h (Fig. 3m). Correspondingly, EdU labelling demonstrates no significant difference in the percentages of cells that are transiting through S-phase at 24 and 48 h in controls (Fig. 3n, p, r) compared with explants treated with SU5402 (Fig. 3o, q, r). Explants treated with SU5402 have a similar surface area when compared with controls at 24 h, and appear smaller at 48 h, although not significantly (Fig. 3s). However, compared with controls (Fig. 3t, v, x), lysotracker staining indicates a significant 2.75- and 3.19-fold increase in apoptosis in explants treated with SU5402 at 24 and 48 h, respectively (Fig. 3u, w, x). Therefore, the attenuation of Fgf signalling and the removal of the AER have similar effects on explant development, and neither affects mesodermal proliferation timing.

**Requirement of Fgf for gene expression**
We sought to determine the global requirement of Fgf signalling during the distal patterning phase, by adding SU5402 to the culture medium, and then performing RNA-sequencing at 48 h on entire

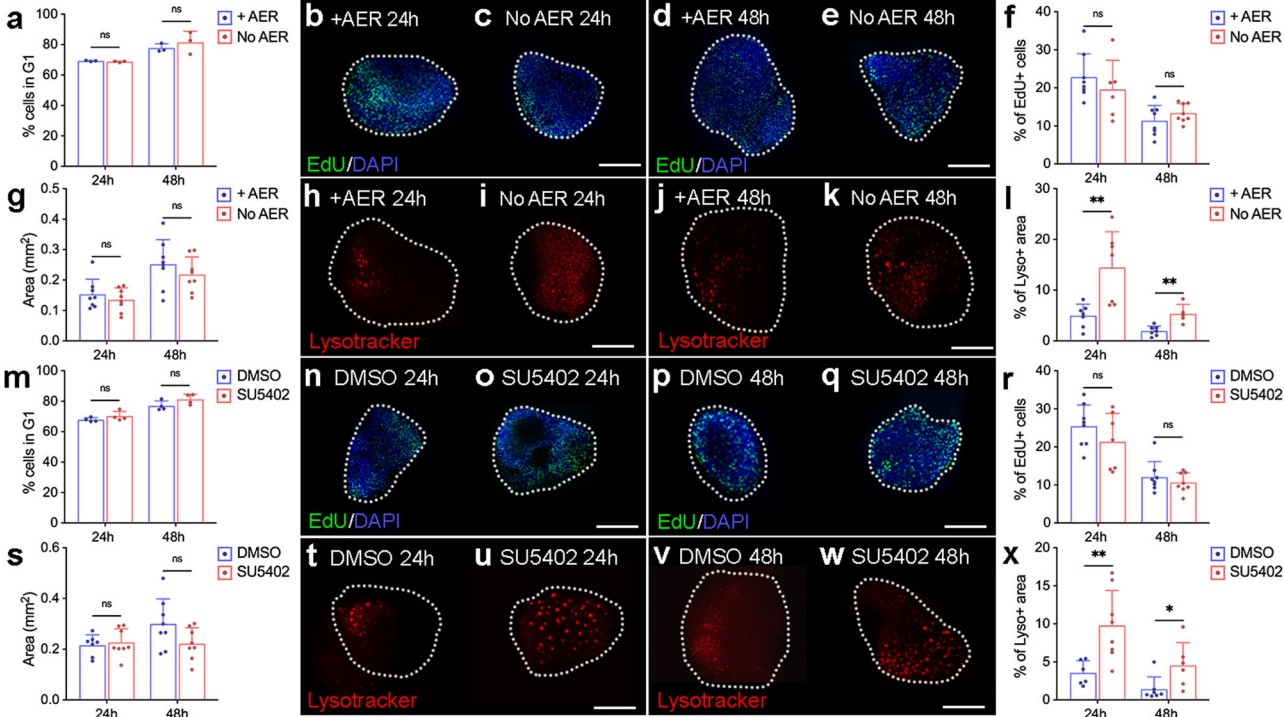

**Fig. 3 | AER removal and Fgf signalling inhibition in explants. a** Flow cytometry reveals no significant difference in G1-phase cells in explants cultured with or without the AER at 24 h ($n = 3,3$, $p = 0.22$) and 48 h ($n = 3,3$, $p = 0.46$). **b, c** EdU labelling in explants cultured with or without the AER at 24 h ($n = 8, 8$) and (**d, e**) at 48 h ($n = 7, 8$). **f** Quantification of EdU labelling shows no significant difference in explants cultured with or without the AER for 24 h ($n = 8, 6, p = 0.39$) and 48 h ($n = 8, 8$ $p = 0.25$). **g** The surface area is not significantly different between explants cultured with or without the AER at 24 h ($n = 8, 8, p = 0.44$) and 48 h ($n = 8, 8, p = 0.35$). **h, i** Lysotracker staining is increased in explants cultured without the AER at 24 h ($n = 7, 7$) and (**j, k**) 48 h ($n = 5, 7$). **l** Lysotracker staining is significantly increased at 24 h ($n = 7, 7, p = 0.005$) and 48 h ($n = 7, 5, p = 0.002$) in explants cultured with an AER 48 h. **m** Flow cytometry reveals no significant difference in G1-phase cells between explants cultured with control DMSO or SU5402 at 24 h ($n = 4, 4, p = 0.21$) and 48 h ($n = 4, 4, p = 0.11$). **n, o** EdU labelling in explants cultured with DMSO or

SU5402 at 24 h ($n = 7, 8$) and (**p, q**) 48 h ($n = 8, 8$). **r** Quantification of EdU labelling shows no significant difference in controls and SU5402-treated explants at 24 h ($n = 8, 7, p = 0.25$) and 48 h ($n = 8, 8, p = 0.42$). **s** The surface area is not significantly different between explants cultured with DMSO or SU5402 at 24 h ($n = 7, 7, p = 0.67$) and 48 h ($n = 7, 7, p = 0.08$). **t, u** Lysotracker staining is increased in explants cultured with SU5402 at 24 h ($n = 6, 8$) and (**v, w**) 48 h ($n = 6, 7$). **x** Lysotracker staining is significantly increased at 24 h ($n = 6, 8, p = 0.009$) and 48 h ($n = 7, 6, p = 0.03$) in explants treated with SU5402. $n = $ biologically independent samples. All bars represent the mean +/− standard deviation represented as error bars. Individual data points are presented as dots overlaid within each bar. Statistical significance was determined through two-tailed unpaired $t$-tests. **$p$-value ≤ 0.01, *$p$-value ≤ 0.05 Scale bars − 200 μM. Source data are provided as a Source Data file.

explants. In total, the expression of 1600 protein coding genes is significantly affected by SU5402 application, of which, 937 are up-regulated and 663 are down-regulated (>2x-fold change with an adjusted $p$-value of <0.05 - Supplementary data 3 – all sequenced genes are shown in Supplementary data 4). We concentrated on genes that have known roles in Fgf signalling and/or limb patterning (Fig. 4a -note log$_2$-fold changes, as well as actual read-counts shown as Fragments per kilobase of transcript per million reads mapped). All four *Sprouty* (*Spry1-4*) genes and *Mkp3*, which are targets of Fgf signalling[43], are down-regulated. *Fgf8* is down-regulated, as are several other genes expressed in the AER, including *Sp8, Hoxc13* and *Dlx5/6*[44]. Consistent with the AER removal experiments, *Shh* is reduced, as are downstream effectors of the signalling pathway, including *Hhip, Ptch1/2* and *Gli1*[45]. *Hoxa11/13/d10/11/12/13* and *Evx2* are down-regulated, with *Hox13* genes showing the greatest reduction in expression. Other down-regulated genes include *Fgf10*, as well as *Bmp2* and its downstream targets, *Msx1* and *Sox9*[46]. Very few genes known to be involved in limb patterning are up-regulated, but exceptions include *Grem1*, which encodes the AER maintenance factor, and *Alx4*, which represses *Shh* expression[47]. HCR in situ hybridisation for *Mkp3, Fgf10, Fgf8, Shh, Hoxa13, Hoxd13, Bmp2* and *Sox9* at 48 h (Fig. 4b–q) confirm the RNA-sequencing data (Fig. 4a). Consistent with the finding that Fgf signalling is dispensable for proliferation (Fig. 3), RNA-sequencing indicates that the expression of the rate-limiting effectors of cell cycle phases are unaffected by the

attenuation of Fgf signalling; *Cyclins D1/2* (*CCND1/2*; G1-phase), *Cyclins E1/E2* (*CCNE1/2*; G1- to S-phase), *Cyclin A2* (*CCNA2*; S- and G2- to M-phase*), Cyclin B2* (*CCNB2*; G2- to M-phase*) (Supplementary Fig. 8a). In addition, *Proliferating Cell Nuclear Antigen* (*PCNA*) expression, which is a reliable indicator of cell cycle progression[48], is unaffected by Fgf signalling inhibition, as determined by RNA-sequencing and HCR in situ hybridisation at 48 h (Supplementary Fig. 8a–c). These observations indicate critical roles for Fgf signalling in activating *Hoxa/d13* expression, and in maintaining the AER and the polarising region in explants. The persistent, albeit reduced expression of *Bmp2* and *Sox9*, suggests that Fgf signalling is dispensable for the onset of chondrogenesis.

## Fgf suppresses myogenesis

The RNA-sequencing data also reveals that the inhibition of Fgf signalling in explants causes a significant up-regulation of genes representing all steps of the myogenic pathway at 48 h (Fig. 5a – Supplementary data 3). These genes include *Pax3* and *Pax7*, which are markers of uncommitted myogenic precursor cells; *Myf5* and *Myod1*, which are involved in myogenic commitment; *Myogenin* (*Myog*), which regulates myogenic induction, and several *Myosin light chain* (*Myl*) and *Myosin heavy chain* (*Myh*) genes, which are involved in myogenic differentiation[49]. *Pax3* expressing myogenic progenitor cells migrate into the limb from the somites[49], and HCR in situ hybridisation reveals

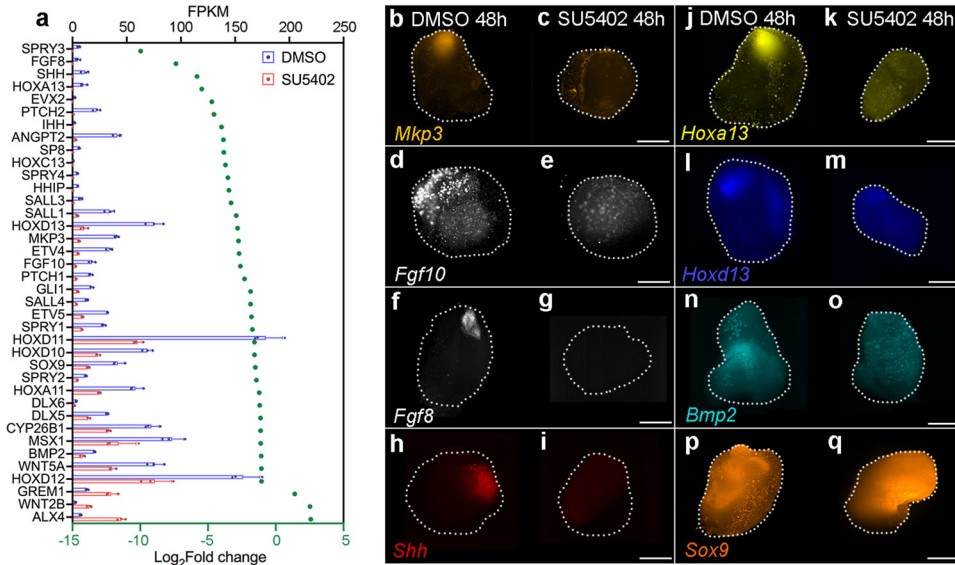

**Fig. 4 | RNA sequencing of SU5402-treated explants. a** Expression of genes involved in Fgf signalling and/or limb development in control DMSO- and SU5402-treated explants at 48 h as shown by log$_2$-fold changes ($n = 3$, >2-fold change; differential expression analysis was performed using DESeq2 (one-tailed Wald test) and the resulting $p$-values were adjusted using the Benjamini and Hochberg procedure for controlling the false discovery rate; adjusted $p$-values of <0.05; green dots) and normalised read-counts mapped to each gene (FKPM – Fragments per kilobase of transcript per million reads mapped). Bars represent the mean +/− standard deviation represented as error bars. Individual data points are presented as dots overlaid within each bar. **b**, **c** *Mkp3*, (**d**, **e**) *Fgf10*, (**f**, **g**) *Fgf8*, (**h**, **I**) *Shh*, (**j**, **k**) *Hoxa13*, (**l**, **m**) *Hoxd13*, (**n**, **o**) *Bmp2*, and (**p**, **q**) *Sox9* expression are down-regulated in SU5402- compared with DMSO-treated explants as shown by HCR in situ hybridisation at 48 h ($n = 4/4$ in each example). Scale bars – 200 μM. $n =$ biologically independent samples. Source data are provided as a Source Data file with individual $p$-values.

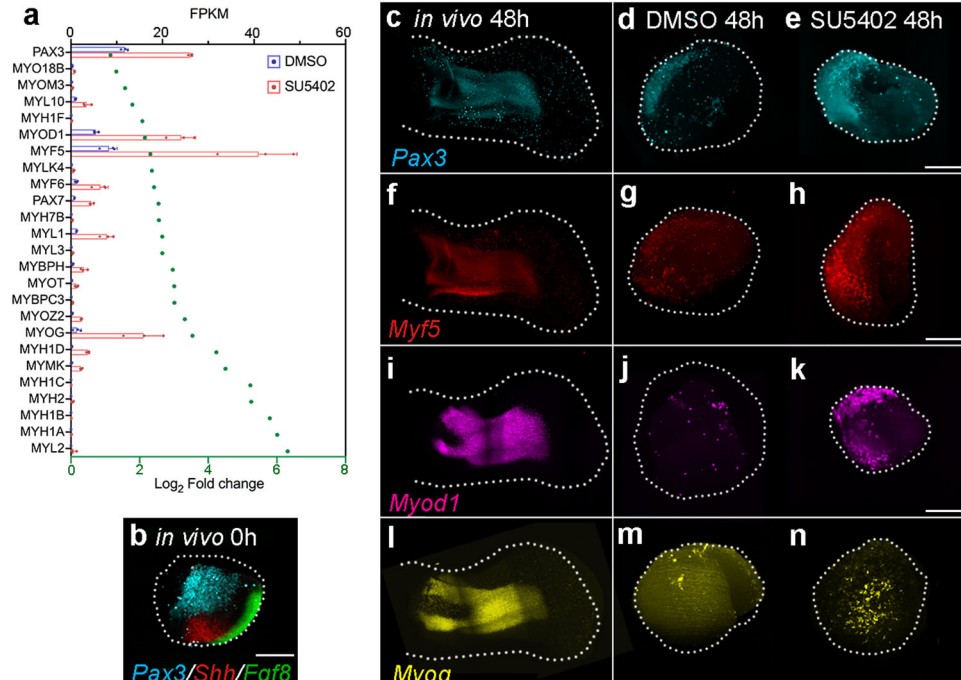

**Fig. 5 | Myogenic gene expression in SU5402-treated explants. a** Expression of genes involved in myogenesis in control DMSO- and SU5402-treated explants at 48 h as shown by log$_2$-fold changes ($n = 3$, >2-fold change; differential expression analysis was performed using DESeq2 (one-tailed Wald test) and the resulting $p$-values were adjusted using the Benjamini and Hochberg procedure for controlling the false discovery rate; adjusted $p$-value of <0.05; green dots) and normalised read-counts mapped to each gene (FKPM – Fragments per kilobase of transcript per million reads mapped; All Bars represent the mean +/− standard deviation represented as error bars. Individual data points are presented as dots overlaid within each bar. **b** *Pax3*-+ve myogenic progenitor cells are present in regions of the wing bud used to make explants as shown by HCR in situ hybridisation at 0 h (*Shh* and *Fgf8* also shown $n = 9/9$). **c** *Pax3*, (**f**) *Myf5*, (**i**) *Myod1* and (**l**) *Myog* expression in wing buds at 48 h ($n = >4$). **d**, **e** *Pax3*, (**g**, **h**) *Myf5*, (**j**, **k**) *Myod1* and (**m**, **n**) *Myog*, are up-regulated in SU5402- compared with DMSO-treated explants at 48 h ($n = 7/7$ in each example). Scale bars – 500 μM in wings; 200 μM in explants. Source data are provided as a Source Data file with individual $p$-values.

a substantial population in tissue dissected at 0 h to make explants, thus explaining why myogenic gene expression is detected (Fig. 5b – *Shh* and *Fgf8* are also shown). *Pax3*, *Myf5*, *Myod1* and *Myog* are restricted to dorsal muscle masses at 48 h in normal wing development (Fig. 5c, f, i, l), and are detectable in control explants (Fig. 5d, g, j, m). However, the attenuation of Fgf signalling causes their significant up-regulation in explants (Fig. 5e, h, k, n), consistent with the RNA-sequencing data (Fig. 5a). These findings show that Fgf signalling suppresses myogenic gene expression in explants.

### Requirement of Fgf for intrinsic mesoderm gene expression

We then examined how the stability of the intrinsic mesoderm transcriptome (Fig. 1) is affected by the attenuation of Fgf signalling in explants (Fig. 4). Analysis of the RNA-sequencing data reveals that only 17% of the genes are significantly down-regulated and that 6% are significantly up-regulated (Fig. 6 - >2x-fold change with an adjusted *p*-value < 0.05). Notably, *Hoxa13*, *Hoxd13* and *Evx2* are among the five most down-regulated genes, the others being *Acot12* and *Sez6l*, which have not been characterised in limb development. Other notable genes that are also moderately down-regulated include *Bmp2* and *Sox9*. These data show that, among known patterning genes that are intrinsically regulated in the mesoderm, *Hoxa/d13* are the most sensitive to the attenuation of Fgf signalling.

## Discussion

We have developed a chick wing tissue explant system and demonstrated that Fgf signalling is required for initiating the distal patterning phase by activating *Hoxa/d13* genes. Following this critical role, Fgf signalling is unexpectedly dispensable for the intrinsic timing of proliferation and differentiation in the mesoderm. In addition to maintaining the AER and the polarising region, Fgf signalling suppresses myogenesis in the limb.

### Limb patterning duration

The antagonism between trunk-derived signals (considered to be RA) and AER-Fgfs controls the early proximal patterning phase, characterised by the expression of *Meis1/2* genes (*Meis*), and the specification of the stylopod and zeugopod[5–10] (Fig. 7). Genetic analyses in the mouse limb indicate that AER-Fgfs induce the expression of the RA-degrading enzyme, *Cyp26b1*[13], whose product creates a gradient of RA signalling, and therefore of Meis[10]. In this model, Meis levels need to be low enough to allow the transition from stylopod to zeugopod specification (Fig. 7). Indeed, the level of RA signalling correlates with—and can change the rate of—5′ *Hox* activation between different avian species[15].

Evidence from both the mouse and chick suggest that the AER-Fgf dependent clearance of RA (Meis) from the distal part of the limb creates a permissive environment that is required for *Hoxa13* expression[6–10], and for the activation of the late distal patterning phase (autopod specification)[16]. Removal of the AER in the chick wing causes the immediate loss of *Hoxa13* expression, which can be restored with an Fgf-soaked bead[50]. Once *Hoxa/d13* genes are activated, the distal mesoderm gains intrinsic properties including a Bmp-dependent proliferation timer, which progressively inhibits G1- to S-phase entry and determines the duration of chick wing patterning[27] (Fig. 7). By developing an ex vivo chick wing tissue explant system, we unexpectedly revealed that distal mesoderm cells maintain normal proliferation/differentiation timing when Fgf signalling is severely curtailed. Under these conditions, our data indicate that *Hoxd13* is activated at sufficient levels to initiate the distal patterning phase in the absence of *Hoxa13*. These observations support genetic analyses in the mouse, demonstrating that distal development is relatively normal in *hoxa13*[-/+]/*hoxd13*[-/-] compound forelimbs (note distal development fails in *hoxa13*[-/-]/*hoxd13*[-/-] limbs)[17]. In addition, although *Hox10/11* genes are down-regulated when Fgf signalling is inhibited in explants,

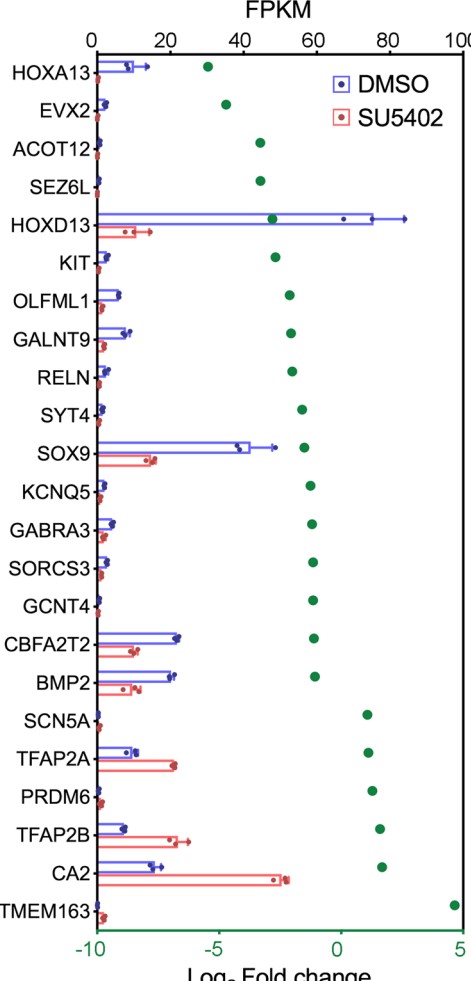

**Fig. 6 | Requirement of Fgf signalling for intrinsic mesodermal gene expression.** Intrinsically activated genes (Fig. 1) that show a significant change in expression when Fgf signalling is attenuated in explants (Fig. 4) as shown by log₂-fold changes ($n = 3$, >2-fold change; differential expression analysis was performed using DESeq2 (one-tailed Wald test) and the resulting *p*-values were adjusted using the Benjamini and Hochberg procedure for controlling the false discovery rate; adjusted *p*-value of <0.05; green dots) and read-counts mapped to each gene (FKPM – Fragments per kilobase of transcript per million reads mapped); Bars represent the mean +/− standard deviation represented as error bars. Individual data points are presented as dots overlaid within each bar. Source data are provided as a Source Data file with individual *p*-values.

they are relatively stable, consistent with their maintained expression following the in vivo removal of the AER in the chick wing[51]. Therefore, Fgf signalling is unexpectedly dispensable for the intrinsic distal programme once it is activated.

The intrinsic mesoderm transcriptome allows us to propose a basic gene regulatory network (GRN) that determines the duration of the late distal patterning phase (Fig. 7). Previous work showed that Hoxa13 directly activates the expression of *Bmp2* and *Bmp7* in distal regions of the mouse limb[52], thus providing a mechanism that coordinates proliferation and differentiation (Fig. 7). We presented evidence that Bmp signalling regulates the Cyclin D-dependant kinase inhibitor, p57[kip2], which could control the decline in the rate of proliferation in the distal mesoderm[27] (Fig. 7). In addition, Bmp signalling induces differentiation by activating primary regulators of chondrogenesis (*Sox9*) and osteogenesis (*Runx2*)[53] (Fig. 7). It is striking that the application of either Bmp2 or Bmp7 partially rescues the phenotype of *hoxa13*[-/-] mutant mouse limbs[52], thus demonstrating the pivotal

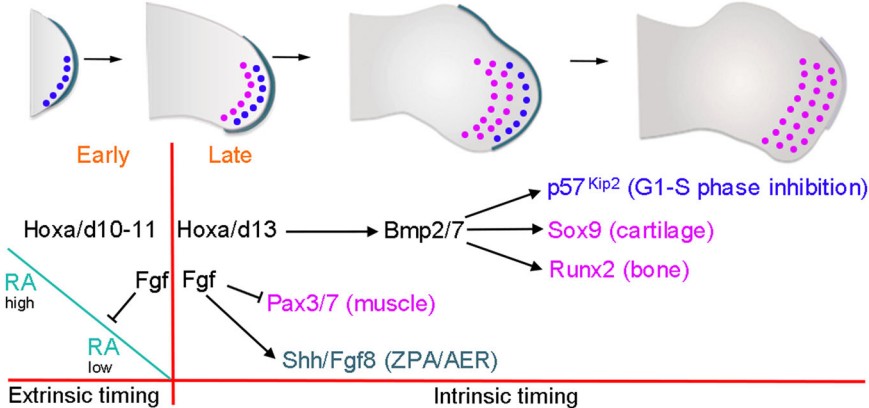

**Fig. 7 | Model of chick wing patterning duration.** Early proximal extrinsic patterning phase - antagonistic flank-derived Retinoic acid (RA) and AER-Fgf signals time *Hoxa/d10/11* gene activation and stylopod and zeugopod positional value specification. Late distal patterning phase - Fgf-depletion creates a permissive environment for the intrinsic activation of *Hoxa/d13* genes and autopod positional value specification. Hoxa/d13 activates *Bmp2/7* independently of Fgf signalling to intrinsically control the timing, hence the duration of mesodermal proliferation, via activation of *p57$^{kip2}$*, which inhibits G1-S-phase progression, and *Sox9/Runx2*, which promote chondrogenesis/osteogenesis. Fgf signalling suppresses myogenesis and maintains the polarising region (ZPA-*Shh*) and the AER (*Fgf8*), which are permissively required for outgrowth.

nature of the Bmp signalling pathway during the late distal patterning phase. Therefore, an implication of our findings is that the AER-Fgf-dependent activation of Hoxa/d13 determines the duration of patterning, by triggering an intrinsic self-terminating process based on the Bmp-dependent timing of proliferation and differentiation (Fig. 7). Another intrinsically regulated gene that is likely to act downstream of Hoxa/d13 encodes the cell adhesion molecule, N-cadherin, which could provide distal mesoderm cells with positional values (e.g., carpal vs. phalange)[37]. It is notable that Bmp signalling progressively increases in the limb[27], and also induces its own inhibitors, such as Grem1[54]. Therefore, it is likely that the dynamics of key effectors of the Bmp signalling pathway determine the duration of the late distal patterning phase.

Our results prompt a re-evaluation of the direct functions of Fgf signalling in limb development. We found that the attenuation of Fgf signalling causes the loss of *Shh* expression in the polarising region, consistent with previous reports[39,40], but also loss of AER-expressed genes including *Fgf8*, and the down-regulation of mesodermal *Fgf10*, which is probably due to the interruption of the reciprocal e-m feedback loop[23,31,32]. These observations could appear surprising because previous work described important roles for Shh signalling in controlling proliferation in the chick wing[38,55]. However, our results here indicate that Fgf signalling is required for supporting the mitogenic functions of Shh signalling. Indeed, the ability of Shh signalling to modulate polarising region proliferation dynamics requires an overlying AER[38]. It is likely that the in vivo function of the AER includes a direct mechanical role that is dispensable in explants. Thus, AER-Fgfs are required for the dorso-ventral flattening of the limb and outgrowth away from the main body[56], possibly by controlling planar cell polarity[57–59] and directional cell division[60]. Loss of such processes could contribute to the excessive apoptosis that occurs when the AER is removed[28] or when AER-Fgf signalling is genetically ablated[30]. Therefore, we have uncovered a minimal proliferation/differentiation timing GRN that can operate without the two principal organisers of limb development - the polarising region (ZPA) and the AER (Fig. 7). One implication is that this GRN represents a core timing mechanism generally used in development, which was hidden because it is normally connected to essential in vivo processes, such as integrated axial patterning.

Our results also suggest that Fgfs are not directly required for the maintenance of an undifferentiated progenitor state in the distal mesoderm, which according to the stable expression of the *Tfap* family of transcription factors in our heterochronic mesoderm grafts, is an intrinsic property. We also revealed that Fgf signalling suppresses the myogenic pathway: a role that we uncovered because normal proliferation/differentiation trajectories are maintained in explants when Fgf signalling is attenuated (Fig. 7). The mutual dependence of epithelial and mesodermal Fgfs makes it difficult to understand which of these signals acts directly to suppress myogenesis. However, early findings, in which the over-expression of *Fgf4* in the chick wing inhibited myogenesis, could indicate an important role for AER-Fgfs[61]. In summary, the explant system uncovers a minimal GRN that coordinates the differentiation of the three primary classes of limb tissue - cartilage, bone and muscle (Fig. 7).

## Perspectives

Parallels can be drawn with the patterning of the limb and of the main body axis. Neuromesodermal progenitors (Nmps) located at the posterior end of the main body transition from producing anterior axial structures - including the vertebrae of the trunk - to producing the posterior tail bud. This involves a switch in gene regulatory activity in which Gdf11 suppresses the anterior programme by inducing the RA-degrading enzyme, *Cyp26a1*[62], to activating the posterior programme via *Hoxb/c13*[63] - a process reminiscent of the function of AER-Fgf signalling in switching proximal to distal limb patterning by antagonising RA signalling via *Cyp26b1*, which allows *Hoxa/d13* activation[13]. During limb development, this switch involves *Hoxa/d* gene transcription directed by structural changes in topologically-associating domains (TADs) located 3′ (up to *Hoxa/d11*) and 5′ (*Hoxa/d13*) to the clusters[64,65]. It is unclear if changes in TAD structure at *Hox* clusters govern the transition from trunk to tail patterning in the main body axis[66], and if this involves a switch from an extrinsic to an intrinsically regulated programme that we describe here for limb development. We speculate that *Hox13* activation triggers a common intrinsic programme that determines the duration of limb and tail patterning. This could be revealed with heterochronic grafting/RNA-sequencing experiments on chick Nmps, like those that we have performed in the limb.

## Methods

This work complies with all ethical regulations because no licensing was required to study chick embryos at the stages used (HH20-HH36).

### Chick husbandry and tissue grafting

Wild type (Henry Stewart, Norfolk, UK) and GFP-expressing (Roslin Institute, Edinburgh, UK) Bovan Brown chicken eggs were incubated and the embryos staged according to Hamburger Hamilton[67].

For tissue grafting, a 150 µm strip of distal mesoderm, including the AER, was dissected from HH20 wing buds. The epithelium was removed after incubation in 0.25% trypsin at room temperature for 2 min and the mesoderm was then cut into 150 µm fragments making cuboid pieces, one of which was placed in a slit between the AER and underlying mesoderm in the mid-distal region of HH24 wing buds using a fine sharpened tungsten needle. All experiments were performed on embryos prior to the determination of sex, which is therefore not relevant to this study.

## Chick wing tissue explants
The posterior third of HH20 wing buds were dissected in ice-cold PBS under a LeicaMZ16F microscope using a fine surgical knife. A bed of 20 µl Growth Factor Reduced Matrigel (Corning) was prepared and allowed to set in four-well plates for 30–40 min at 37 °C. The explants were then placed on top of the Matrigel (4–5 per well), covered with another layer of Matrigel, and cultured in CMRL media supplemented with 10% FBS/1% Pen Strep/1% L-Glut in a humidified incubator with 5% $CO_2$ at 37 °C. Explants were collected by replacing the culture media with Cell Recovery Solution (Corning) on ice for 1 h.

## AER removal
The AER of wing buds of HH20 embryos was stained with 1% Nile blue solution in ovo and removed by pulling with sharp forceps. Embryos were then collected, and the right-hand wing bud dissected to make explants as described above (the left-hand wing bud was used to make control explants).

## SU5402 treatment
SU5402 (Sigma) dissolved in DMSO with a final concentration of 5 µM was added to the explant culture media at 0 h. DMSO only was used as a control.

## Explant size measurements
Explants were placed in a Petri dish containing 1XPBS and imaged using a LeicaMZ16F microscope and LAS X 1.1.0.12420 imaging software. The surface areas of the explants were measured using the Record Measurement Feature of the Lasso tool in Adobe Photoshop 2020.

## RNA-fluorescence in situ hybridisation with amplification by hybridisation chain reaction
Samples were fixed in 4% PFA at 4 °C overnight, then washed in PBS and progressively dehydrated through a methanol series and stored in methanol at −20 °C. The samples were then rehydrated in a methanol to PBT series and treated with proteinase K for 5–7 min for explants and 15–20 min for wing buds, followed by post-fixing in 4% PFA for 20 min. At this point an optional bleaching protocol described below was performed. Samples were further washed with PBT, 5X SSCT and Molecular Instruments Probe hybridisation buffer before addition of the probe overnight at 37 °C. All probes were custom generated (Molecular Instruments Inc.), and accession numbers are provided in Supplementary data 5. The probes were prepared by adding 8 µl of 1 µM probe to 500 µl of Molecular Instruments Probe hybridisation buffer. The next day samples were washed with Molecular Instruments Probe Wash Buffer, 5X SSCT and Molecular Instruments Amplification buffer. Amplifier pairs were prepared by heat-shocking at 95 °C for 90 s and snap cooling for 30 min in the dark at room temperature, before being added to the samples in the Molecular Instruments Amplification buffer. The samples with amplifiers in Molecular Instruments Amplification buffer were incubated overnight in the dark at room temperature. The next day samples were washed and stored in 5X SSCT before imaging. Fluorescent images of HCRs were taken with a Zeiss Z1 Lightsheet Microscope with a 10X objective and Zen Black 2014SP1 imaging software. Images were processed with ImageJ v2.14.0

(FIJI) and Adobe Photoshop 2020. Limb bud images in Fig. 7 were taken as tiled images and stitched together with the Grid/Collection stitching plugin in ImageJ v2.14.0 (FIJI)[68]. Autofluorescence bleaching was used for HCR on the limb buds in Fig. 5. Samples were washed with PBT followed by incubation in a 3% $H_2O_2$ 20 mM NaOH PBT solution on ice for 3 h to reduce sample autofluorescence. The samples were then washed in PBT for 3 × 10 min 3 before proceeding with HCR.

## Whole mount in situ hybridisation
Embryos were fixed in 4% PFA overnight at 4 °C, dehydrated in methanol overnight at −20 °C, rehydrated through a methanol/PBS series, washed in PBS, then treated with proteinase K for 20 min (10 µg/ml⁻¹), washed in PBS, fixed for 30 min in 4% PFA at room temperature and then prehybridised at 69 °C for 2 h (50% formamide/50% 2x SSC). 1 µg of antisense DIG-labelled mRNA probes were added in 1 ml of hybridisation buffer (50% formamide/50% 2x SSC) at 69 °C overnight. Embryos were washed twice in hybridisation buffer, twice in 50:50 hybridisation buffer and MAB buffer, and then twice in MAB buffer, before being transferred to blocking buffer (2% blocking reagent 20% lamb serum in MAB buffer) for 2 h at room temperature. Embryos were transferred to blocking buffer containing anti-digoxigenin antibody (1:2000) at 4 °C overnight, then washed in MAB buffer overnight before being transferred to NTM buffer containing NBT/BCIP and mRNA distribution visualised using a LeicaMZ16F microscope and LAS X 1.1.0.12420 imaging software.

## Bead implantation
Affigel beads (Biorad) were soaked in human Bmp2 protein (0.05 µg/µl¹ - R&D) dissolved in PBS/4 mM HCl or Noggin protein (0.05 µg/µl¹ - R&D) dissolved in PBS/4 mM HCl. Beads were soaked for 2 h and implanted into distal mesoderm using a sharp needle.

## Flow cytometry for cell cycle analyses
Explants and equivalent regions of stage-matched distal mesoderm were dissected in ice cold PBS under a LeicaMZ16F microscope using a fine surgical knife and pooled from replicate experiments ($n = 10$–12), before being digested into single cell suspensions with trypsin (0.05%, Gibco) for 30 mins at room temperature. Cells were briefly washed twice in PBS, fixed in 70% ethanol overnight, washed in PBS and re-suspended in PBS containing 0.1% Triton X-100, 50 µg/ml⁻¹ of propidium iodide and 50 µg/ml⁻¹ of RNase A (Sigma). Dissociated cells were left at room temperature for 20 min, cell aggregates were removed by filtration and single cells analysed for DNA content with a FACSCalibur flow cytometer using BD CellQuest™ Pro Software (BD Biosciences). Single cells were gated using the forward scatter to determine which cells were doublets and therefore excluded in the gate. Raw data is included in Supplementary data 6 showing the gating strategy. Based on ploidy values, cells were assigned in G1, S, or G2/M phases and this was expressed as a percentage of the total cell number (5000–12,000 cells in each case). Statistical significance of numbers of cells in different phases of the cell cycle (G1 vs. S, G2 and M) between pools of dissected wing bud tissue and explants with $n > 3$ was determined by two-tailed unpaired $t$-tests to obtain $p$-values (significantly different being a $p$-value of less than 0.05).

## EdU labelling
Explants were incubated in 0.5 mM EdU in CMRL for 2 h at 37 °C, then fixed with 4% PFA for 15 min and washed with 3% BSA/PBS at room temperature, before being permeabilised with 0.5% Triton X-100 in PBS for 20 min. Explants were incubated in the dark for 1 h in a click-kit reaction cocktail containing Azide Dye (Molecular Probes) and washed with 3% BSA/PBS and counterstained with DAPI (1:1000 in 3% BSA/PBS) for 10 min, followed by 3 washes in 3% BSA/PBS before imaging. Fluorescent images of EdU/DAPI labelled explants were taken with a

Zeiss Apotome 2 microscope using a 10X objective and Axiovision software (Zeiss). Quantification of the percentage of the EdU positive area (encompassing the entire explant) compared to the DAPI area was performed using ImageJ v 2.14.0 (FIJI). Briefly, binary masks of the DAPI and EdU positive area in the Z-stacks (30–40 slices per explant) were created using the Moments Auto Threshold function. The Analyse Particles command was then used to measure the area of DAPI in the Z-stack. The area of EdU within the area of DAPI was measured by applying the Analyse Particles command in the EdU mask.

### Apoptosis assays

Explants were transferred to a Lysotracker (Life Technologies, L-7528)/PBS solution (1:1000) in the dark, incubated for 1 h at 37 °C, washed in PBS, and fixed overnight in 4% PFA at 4 °C. Explants were then washed in PBS and progressively dehydrated in a methanol series before imaging. Fluorescent images of Lysotracker labelled explants were taken with Zeiss Apotome 2 microscope using a 10X objective and Axiovision software (Zeiss). Images were processed using ImageJ v2.14.0 (FIJI). The Lysotracker positive area was measured from maximum intensity projection images using the Limit to Threshold Measure feature combined with a manually selected threshold.

### RNA sequencing analyses and clustering

RNA sequencing was performed on two conditions: grafts as described above with equivalent in vivo tissue from the contralateral wing, and on explants treated with either DMSO or SU5402. Three replicate experiments were performed for each condition ($n = 10$–12 tissue samples in each experiment). Samples were collected by flash freezing in dry ice. Total RNA was extracted from samples using Trizol-chloroform extractions. Explant RNA was further concentrated using Zymo RNA clean and concentrator kit as per manufacturer's instructions. Messenger RNA was purified from total RNA using poly-T oligo attached magnetic beads for explants. Sequencing was performed in either Illumina HiSeq 2000 PE50 (Grafts) or Illumina NovaSeq 6000 PE150 (Explants). Sequencing data were mapped using HISAT v2.0.3 (Grafts) or v2.0.5 (Explants) to the chicken reference genome. Based on quality control checks one of the HH24 and HH24g samples was excluded from further analysis. Quantification of gene expression level for explants was performed with Feature Counts v1.5.0-p3 and then FPKM (Fragments per kilobase of transcript per million reads mapped) of each gene was calculated based on the length of the gene and read counts mapped to this gene. Differential expression analysis between DMSO and SU5402 samples was performed using DESEq2 v1.20.0[69] (one-tailed Wald test - resulting $p$-values were adjusted using the Benjamini and Hochberg procedure for controlling the false discovery rate). Genes with an adjusted $p$-value < 0.05 and a fold-change >2 were assigned as differentially expressed. For grafts, the count data for the samples were normalised using trimmed mean of $m$-value normalisation and transformed with Voom, resulting in $\log_2$-counts per million with associated precision weights. A heat-map was made showing the correlation (Pearson) of the normalised data collapsed to the mean expression per group. A statistical analysis using an adjusted $p$-value < 0.05 and a fold-change >2 identified differentially expressed genes in the two contrasts evaluated. Gene clusters were identified from the set of differentially expressed genes. The evaluation considered between two and 35 clusters using hierarchical, $k$-means, and PAM clustering methods based on the internal, stability and biological metrics provided from the clValid R package. Most of the internal validation and stability metrics indicated that either the lowest possible number or conversely the highest number evaluated were preferable. The metrics giving more nuanced information in the intermediate range were the Silhouette measure, and the Biological Homogeneity Index (BHI). Based on manual inspection it was decided to use hierarchical clustering with $k = 9$ gene clusters, which showed favourable properties for both these measures.

### Alcian blue skeletal preparations

Embryos were fixed in 90% ethanol for 2 days then transferred to 0.1% Alcian blue in 80% ethanol/20% acetic acid for 1 day, before being cleared in 1% KOH.

### Statistics & reproducibility

All multiplexed hybridisation chain reaction (HCR), EdU labelling and Lysotracker assays were performed on over 3 biological replicate explants (individual $n$-numbers are provided in respective figure legends and data is available in the Source Data file). All attempts at replication of these experiments were successful and we have included representative images with replicate information in the manuscript. To determine statistical significance in EdU labelling and Lysotracker experiments, two-tailed Student's $t$-tests were used. To determine statistical significance of numbers of cells in different phases of the cell cycle (G1 vs. S, G2 and M) between pools of 10-12 explants in flow cytometry experiments, two-tailed Student's $t$-tests were used. In all cases significantly different is taken as a $p$-value of less than 0.05 and GraphPad Prism9 was used to construct graphs. Gating information for flow cytometry is found in Supplementary data 6. No statistical method was used to predetermine sample size and no data were excluded from the analyses. The experiments were not randomised, and the Investigators were not blinded to allocation during experiments and outcome assessment.

### Reporting summary

Further information on research design is available in the Nature Portfolio Reporting Summary linked to this article.

## Data availability

The datasets generated and/or analysed during the current study are publicly available. The RNA sequencing data is publicly available in GEO for explants in GSE223444 and for grafts in GSE232959. All flow cytometry data generated or analysed during this study is included in this published article (and in Supplementary data 6). All other data associated with figures in this study is included in the Source Data file. Source data are provided with this paper.

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

## Acknowledgements

This work was supported by the Wellcome Trust (202756/Z/16/Z) to M.T. and (212247/Z/18/Z) to M.P. and by the Spanish Ministry of Science and Innovation (Grant PID2020-114525GB-I00) to M.R.

## Author contributions

S.S.P., C.M., H.S. did the explant work. K.C. did the in vivo grafting experiments and J.P. processed the samples. M.P. assisted with explant protocol development. P.S-L. and M.R. contributed to the in vivo grafting experiments. M.T. designed the project wrote the paper and did the in vivo bead experiments. All authors edited the manuscript.

## Competing interests

The authors declare no competing interests
