## [Peer Review File · Nature Communications]

Fgf signalling triggers an intrinsic mesodermal timer that determines the duration of limb patterningREVIEWER COMMENTS

Reviewer #1 (Remarks to the Author):

This paper concerns the role of FGF signalling (ostensibly from the AER- but see point 2 below) in chick limb development. There are three "take home lessons" : 1) defining the transcriptome of the chick wing bud 2) developing an explant system for development of the embryonic chick wing 3) analysis of FGF signaling in this explant system via two manipulations- removal of the AER and pharmacological inhibition of FGF signaling by SU5402. The authors find that FGF signalling is dispensable for proliferation, as well as maintenance of the intrinsic transcriptome. However, they find it is required to trigger this program, possibly by activating Hox13 gene.

1) The authors should provide a more complete description of their experiment, that generated the data in Figure 1, that its definition of the intrinsic mesoderm transcriptome is the bedrock of the rest of the paper. Specifically...

a) In regard to their HH24g data (of 150 um of the distal-most grafted mesoderm), how much is host and graft? In trying to think about this, I note in the M&M, "...the mesoderm was then cut into cubes, which were placed in slits between the AER and underlying mesoderm in the mid-distal region of HH24 wing buds. How many cubes are inserted in the HH24 hosts? From their cartoon in Figure 1B, there looks like one relatively small GFP+ domain. Is this the case? Then most of their RNA data would from the host cells.

b) In their description of this experiment, on page 5, they describe RNA-sequencing on grafted mesoderm and "and for controls, equivalent (non-grafted) tissue in the contralateral wing bud.". But the data in Figure 1D indicates another control: HH24 limbs. They should mention this also.

c) They should describe where the GFP expression is coming from in the donor tissue.

2) Given the importance of the reciprocal regulation loop between FGF8 and FGF10, the authors should show data regarding Fgf10 expression in their AER removal experiments as well as the SU4052 addition experiments. It is possible that any conclusions the authors reach regarding an FGF effect, may be due to a loss of Fgf10 signaling.

3) The authors show that Mkp3 expression is downregulated when the AER is removed (Fig 4c and d). On page 8 they contend that this downregulation "indicates that signalling by mesodermal Fgfs (including Fgf10 that reciprocally maintains Fgf8 activity) is also affected by the removal of the AER (44). This is very confusing. They seem to be implying that the loss of the AER is known to affect Mkp3 expression indirectly, through its effect on Fgf10 expression. As far as I know, this is not the case and is not the case in reference 44. Also, what other mesodermal Fgfs, other than Fgf10, are the authors referring to?

Reviewer #2 (Remarks to the Author):

In this study, authors utilized wing explant culture system to dissect the roles of AER-Fgf signaling on the distal patterning. By using this system, they showed that Fgfs are required for activation of Hox13 genes, but not required thereafter for mesodermal proliferation. They also showed that the inhibition of Fgf signaling in explants enhances myogenic gene expression. Based on their results, authors proposed a model that Fgf signaling triggers an intrinsic mesodermal timer that determines the duration of limb patterning.

Their finding that Fgfs are dispensable for mesodermal proliferation in wing explants is interesting, but data provided is too preliminary to draw any conclusions.

1. The main finding of this manuscript is that Fgfs are dispensable for mesodermal proliferation at least in wing explants (Figure 4~). Fig 1 has little to do with the main finding, and Figs 2-3 are just

examinations of explant culture conditions. Thus, I suggest authors to move these figures to the Supplemental figures.

2. Figure 4: EdU labeling and LysoTracker staining should be performed at 48h after removal of the AER and treatment with SU5402, as authors performed flow cytometry and RNA sequencing at 48h.

3. Figure 4: Statistic data should be provided regarding the number of EdU- and LysoTracker-positive cells both at 24 h and 48h.

4. Figure 4: RNA sequencing data of genes involved in regulation of cell cycle controls and proliferation (including genes of each cell cycle phase) of the distal explants treated with SU5402 +48h should be provided. If Fgf signaling is dispensable for proliferation, we expect to see the unaltered expression levels of these genes. This should not be difficult as RNA sequencing of these samples has already been performed (Figure 5).

5. Figure 5, Figure 6, Figure 7: It is puzzling that the authors describe the changes in expression levels of Bmp2 and Sox9 as "SLIGHTLY reduced" when they are reduced by half (Figure 5, 6), and as "SIGNIFICANTLY up-regulated" when the expression level of Pax3 is doubled (Figure 7). It is necessary to show statistic data whether the changes in expression levels are significant, at the very least, for the genes whose expression levels are being discussed.

6. Figure 7: It is well known fact that inhibition of Fgf signaling activates myogenesis (reviewed by Pawlikowski et al., 2017 Dev Dyn), and the provided data add little to what is already known. I suggest authors to move this figure to Supplemental figure.

7. Figure 8: It is a leap of logic to assume that endogenous Bmp regulates p57kip2 expression, which in turn regulates the proliferation rate of distal mesoderm, based on the (exogenous) Bmp-soaked bead implantation results (Figure S4). Since the role of Bmp and p57kip2 on the proliferation rate is critical for the proposed model, more direct evidence should be provided. I suggest authors to suppress endogenous Bmp level by using Noggin (and confirming it with decreased Msx expression level) and examine the expression level of p57kip2. I also suggest to see whether reduction of p57kip2 function affects the proliferation levels in the distal mesenchyme of wing buds.

1) The authors should provide a more complete description of their experiment, that generated the data in Figure 1, that its definition of the intrinsic mesoderm transcriptome is the bedrock of the rest of the paper. Specifically...

a) In regard to their HH24g data (of 150 μ m of the distal-most grafted mesoderm), how much is host and graft? In trying to think about this, I note in the M&M, "...the mesoderm was then cut into cubes, which were placed in slits between the AER and underlying mesoderm in the mid-distal region of HH24 wing buds. How many cubes are inserted in the HH24 hosts? From their cartoon in Figure 1B, there looks like one relatively small GFP+ domain. Is this the case? Then most of their RNA data would from the host cells.

We have improved the clarity of this section by indicating that we dissected a strip of distal mesoderm measuring 150 μ m thick, excluding the ZPA. Then, the ectoderm was removed through mild trypsin digestion. Finally, the mesoderm strip was fragmented into approximately 150 μ m-sized cuboid pieces, one of which was grafted under the AER of HH24 host embryos. Only GFP+ve donor tissue was used for the RNA-sequencing, and we have depicted this in Figure 1a.

b) In their description of this experiment, on page 5, they describe RNA-sequencing on grafted mesoderm and "and for controls, equivalent (non-grafted) tissue in the contralateral wing bud.". But the data in Figure 1D indicates another control: HH24 limbs. They should mention this also.

Thank you for noticing this, we have now included this point.

c) They should describe where the GFP expression is coming from in the donor tissue.

We have mentioned that GFP is ubiquitously and constitutively produced in the GFP+ve transgenic chicks that we used for donors.

2) Given the importance of the reciprocal regulation loop between FGF8 and FGF10, the authors should show data regarding *Fgf10* expression in their AER removal experiments as well as the SU4052 addition experiments. It is possible that any conclusions the authors reach regarding an FGF effect, may be due to a loss of *Fgf10* signaling.

We have now included this data as Supplementary Figures 6 and 7. We find, as expected from the reciprocal nature of the FGF8-FGF10 feedback loop, that the expression of *Fgf10* is down-regulated when the AER is removed or when SU5402 is applied (fitting with the RNA-seq data – HCR of *Fgf10* at 48h is now also included in Figure 4). This is totally consistent with the reported *Fgf10* downregulation observed in the distal mesoderm after loss of *Fgf8* and *Fgf4* (Sun, Nature 2002; Boulet, Development 2004).

As the Reviewer correctly points out, such considerations could make it difficult to assign specific functions to either AER-FGFs (in particular FGF4 and FGF8) or mesodermal-FGFs (in particular FGF10). However, the major roles that we describe for FGF signalling are in the activation of *Hoxa/d13* expression, and in repressing genes involved in myogenesis. A wealth of experimental data, in particular from the Miguel Torres lab, has shown that the antagonism between RA from the flank and AER-FGF signalling is required for *Hoxa/d13* activation. In addition, AER-FGF4 signalling was found to down-regulate myogenic gene expression (Edam-Vovard, Dev Biol, 2001). Therefore, we are confident that the roles that we describe for FGFs involve AER-FGFs. However, the most important results in our paper describe processes that occur normally without either AER-FGFs or mesodermal-FGFs.

3) The authors show that *Mkp3* expression is downregulated when the AER is removed (Fig 4c and d). On page 8 they contend that this downregulation "indicates that signalling by mesodermal *Fgfs* (including *Fgf10* that reciprocally maintains *Fgf8* activity) is also affected by the removal of the AER (44). This is very confusing. They seem to be implying that the

loss of the AER is known to affect *Mkp3* expression indirectly, through its effect on *Fgf10* expression. As far as I know, this is not the case and is not the case in reference 44. Also, what other mesodermal Fgfs, other than *Fgf10*, are the authors referring to?

Since FGF8 and FGF10 operate in a reciprocal feedback loop, it is expected that signalling by both proteins will be affected when the AER is removed. Our new data indicates that FGF8 plays a predominant role in signalling to the mesoderm, since the removal of the AER only slightly down-regulates *Fgf10* expression, but dramatically down-regulates *Mkp3* expression. In reference 44 (now 23) the authors described the interaction between FGF8 and FGF10. However, this was several years before *Mkp3* was identified as a target of FGF signalling. We have included these points in our revised manuscript, and we have also indicated that when we consider mesodermal FGFs, we are referring, in particular to, FGF10. '.... signalling by mesodermal Fgfs is also affected by the removal of the AER - in particular, *Fgf10*, that reciprocally maintains *Fgf8* activity'.

Reviewer #2 (Remarks to the Author):

In this study, authors utilized wing explant culture system to dissect the roles of AER-Fgf signaling on the distal patterning. By using this system, they showed that Fgfs are required for activation of *Hox13* genes, but not required thereafter for mesodermal proliferation. They also showed that the inhibition of Fgf signaling in explants enhances myogenic gene expression. Based on their results, authors proposed a model that Fgf signaling triggers an intrinsic mesodermal timer that determines the duration of limb patterning. Their finding that Fgfs are dispensable for mesodermal proliferation in wing explants is interesting, but data provided is too preliminary to draw any conclusions.

1. The main finding of this manuscript is that Fgfs are dispensable for mesodermal proliferation at least in wing explants (Figure 4~). Fig 1 has little to do with the main finding, and Figs 2-3 are just examinations of explant culture conditions. Thus, I suggest authors to move these figures to the Supplemental figures.

We feel that Figure 1 is important because it describes the identification of the intrinsic distal mesoderm transcriptome, which allows us to draw the main conclusions regarding its regulation by FGF signalling in explants (New Figure 6). As suggested, we have moved the description of the explant culture conditions to the Supplementary data (Figure S5). We would like to keep the gene expression data as a main figure (Figure 2), because detailed HCR patterns have not been documented before in limbs or in explants.

2. Figure 4: EdU labeling and LysoTracker staining should be performed at 48h after removal of the AER and treatment with SU5402, as authors performed flow cytometry and RNA sequencing at 48h.

We have done these experiments and included the data in the new Figure 3.

3. Figure 4: Statistic data should be provided regarding the number of EdU- and LysoTracker-positive cells both at 24 h and 48h.

We have included this data in the new Figure 3. We had to repeat all of the EdU experiments, because in order to count cells, we had to co-stain with DAPI. We find no significant difference between the numbers of EdU+ve cells when the AER is removed or when SU5402 is applied. We also repeated the lysoTracker staining as we ran them in parallel with the EdU experiments for consistency. We measured the lysoTracker+ve area in explants as we were unable to accurately count cells because they are undergoing lysis. We

found a significant increase in apoptosis when the AER is removed or when SU5402 is applied.

4. Figure 4: RNA sequencing data of genes involved in regulation of cell cycle controls and proliferation (including genes of each cell cycle phase) of the distal explants treated with SU5402 +48h should be provided. If Fgf signaling is dispensable for proliferation, we expect to see the unaltered expression levels of these genes. This should not be difficult as RNA sequencing of these samples has already been performed (Figure 5).

We have included this new data as Supplementary Figure 8a. We have included all of the Cyclins found in our RNA-seq dataset that are rate limiting for progression through the different cell cycle phases. We find that the expression of none of them is affected by the attenuation of FGF signalling. We also find that the expression of *PCNA* is unaffected, which is an accurate indicator of cell cycle progression. We have also included HCR *in situ* data showing that the expression of *PCNA* is unaltered by the attenuation of FGF signalling in explants at 48h (Supplementary Figure 8b).

5. Figure 5, Figure 6, Figure 7: It is puzzling that the authors describe the changes in expression levels of *Bmp2* and *Sox9* as “SLIGHTLY reduced” when they are reduced by half (Figure 5, 6), and as “SIGNIFICANTLY up-regulated” when the expression level of *Pax3* is doubled (Figure 7). It is necessary to show statistic data whether the changes in expression levels are significant, at the very least, for the genes whose expression levels are being discussed.

We apologise for our lack of consistency. We have removed instances in which we have verbally described changes in gene expression (i.e., moderately etc), other than to say that a gene is down- or up-regulated at significant levels according to the statistics. For quantification of the data, and to allow readers to assess genes expression changes for themselves, the \log_2 -fold changes (significant is >2-fold change; adjusted *p*-value of <0.05) and normalised read-counts that are mapped to each gene, are shown in the Figures.

6. Figure 7: It is well known fact that inhibition of Fgf signaling activates myogenesis (reviewed by Pawlikowski et al., 2017 Dev Dyn), and the provided data add little to what is already known. I suggest authors to move this figure to Supplemental figure.

We would prefer to keep this data as a main figure. Although FGF signalling has been shown to be involved in myogenesis in other contexts, its role in limb development is less clear because of its essential *in vivo* requirement.

7. Figure 8: It is a leap of logic to assume that endogenous *Bmp* regulates *p57kip2* expression, which in turn regulates the proliferation rate of distal mesoderm, based on the (exogenous) *Bmp*-soaked bead implantation results (Figure S4). Since the role of *Bmp* and *p57kip2* on the proliferation rate is critical for the proposed model, more direct evidence should be provided. I suggest authors to suppress endogenous *Bmp* level by using *Noggin* (and confirming it with decreased *Msx* expression level) and examine the expression level of *p57kip2*. I also suggest to see whether reduction of *p57kip2* function affects the proliferation levels in the distal mesenchyme of wing buds.

We have done these experiments and included the data in Supplementary Figure 4. We find that *Bmp2* supplied on beads increases *p57kip2* and *Msx2* expression, and that *Noggin* decreases *p57kip2* and *Msx2* expression. We feel that it would be a huge amount of work to attempt to attenuate *p57kip2* function *in vivo*. Since *p57kip2* has a specific role in inhibiting proliferation that has been well-documented in many contexts, we feel this would not add any additional insight to the main conclusions of the paper.

REVIEWERS' COMMENTS

Reviewer #1 (Remarks to the Author):

The authors have addressed all of my questions and concerns -
Very interesting work that deserves publication in this journal.

Mark Lewandoski

Reviewer #2 (Remarks to the Author):

The authors have addressed each issue that I raised, and thus I am happy to recommend this article for publication in Nature Communications.